# Pseudomonas Exotoxin A Based Toxins Targeting Epidermal Growth Factor Receptor for the Treatment of Prostate Cancer

**DOI:** 10.3390/toxins12120753

**Published:** 2020-11-28

**Authors:** Alexandra Fischer, Isis Wolf, Hendrik Fuchs, Anie Priscilla Masilamani, Philipp Wolf

**Affiliations:** 1Faculty of Medicine, University of Freiburg, 79106 Freiburg, Germany; alexandra.fischer.uro@uniklinik-freiburg.de (A.F.); isis.wolf@uniklinik-freiburg.de (I.W.); anie.priscilla.masilamani@uniklinik-freiburg.de (A.P.M.); 2Department of Urology, Antibody-Based Diagnostics and Therapies, Medical Center—University of Freiburg, Breisacher Str. 66, 79106 Freiburg, Germany; 3Institute of Laboratory Medicine, Clinical Chemistry and Pathobiochemistry, Charité—Universitätsmedizin Berlin, corporate member of Freie Universität Berlin, Humboldt-Universität zu Berlin, and Berlin Institute of Health, 13353 Berlin, Germany; hendrik.fuchs@charite.de

**Keywords:** prostate cancer, targeted toxins, epidermal growth factor, epidermal growth factor receptor, *Pseudomonas* Exotoxin A

## Abstract

The epidermal growth factor receptor (EGFR) was found to be a valuable target on prostate cancer (PCa) cells. However, EGFR inhibitors mostly failed in clinical studies with patients suffering from PCa. We therefore tested the targeted toxins EGF-PE40 and EGF-PE24mut consisting of the natural ligand EGF as binding domain and PE40, the natural toxin domain of *Pseudomonas* Exotoxin A, or PE24mut, the de-immunized variant thereof, as toxin domains. Both targeted toxins were expressed in the periplasm of *E.coli* and evoked an inhibition of protein biosynthesis in EGFR-expressing PCa cells. Concentration- and time-dependent killing of PCa cells was found with IC_50_ values after 48 and 72 h in the low nanomolar or picomolar range based on the induction of apoptosis. EGF-PE24mut was found to be about 11- to 120-fold less toxic than EGF-PE40. Both targeted toxins were more than 600 to 140,000-fold more cytotoxic than the EGFR inhibitor erlotinib. Due to their high and specific cytotoxicity, the EGF-based targeted toxins EGF-PE40 and EGF-PE24mut represent promising candidates for the future treatment of PCa.

## 1. Introduction

Prostate cancer (PCa) is the second most common malignancy in men worldwide. More than 1.27 million new cases and more than 358,000 deaths are expected from this tumor every year [1]. Primary tumors can be successfully treated by surgery or local radiation. However, despite improved therapeutic options, such as androgen deprivation therapy, radiation, and chemotherapy, curative treatment is no longer possible, once the tumor has spread [2].

In recent years, targeted therapy has been established as a new cornerstone beside the classical treatment options for advanced PCa [3,4,5]. In the search for antigens that could serve as targets, the focus, among the prostate specific membrane antigen (PSMA) [5,6] or the prostate stem cell antigen (PSCA) [7], has been on the epidermal growth factor receptor (EGFR) [8,9,10]. EGFR belongs to the ErbB receptor tyrosine kinase family [11]. It is a 1186 amino acid transmembrane glycoprotein comprised of an *N*-terminal 621 amino acid (aa) extracellular domain, a 23 aa transmembrane domain, and a 542 aa cytoplasmic domain including tyrosine kinase activity and *C*-terminal phosphorylation sites [12,13]. Seven ligands were described to bind to EGFR: the epidermal growth factor (EGF), the transforming growth factor α (TGFα), amphiregulin, betacellulin, epigen, epiregulin, and the heparin binding EGF-like growth factor [14]. After ligand binding, EGFR homodimerizes or heterodimerizes with other members of the ErbB family (HER2/ErbB2, ErbB3, or ErbB4) followed by autophosphorylation of the intracellular tyrosine kinase domain and activation of signaling pathways associated with cell proliferation, growth, differentiation, migration, and apoptosis inhibition [15].

EGFR signaling was found to play a major role in the tumorigenesis of PCa in view of proliferation, survival, invasiveness, and metastasis [16,17,18,19]. In different studies, EGFR expression was found in 18–75.9% of patients with prostate adenocarcinoma and in 100% of patients with hormone-refractory metastatic disease [20,21,22]. EGFR overexpression was significantly associated with Gleason score, recurrence, castration resistant disease, and poorer disease-free survival [20,21,22]. Moreover, EGFR mediates docetaxel resistance in human castration-resistant PCa through the Akt-dependent expression of ABCB1 (MDR1) [23].

Different inhibitors against EGFR, like Erlotinib, Gefitinib, Vandetanib (which additionally inhibits VEGF), and Lapatinib (which additionally inhibits HER2) were tested alone or in combination with chemotherapy in phase II studies on patients with castration resistant PCa [24,25,26,27]. However, results were disappointing and no or only low anti-cancer activity in some patients could be registered. Treatment of patients with the anti-EGFR mAb Cetuximab plus Docetaxel were more promising. In 20% and 31% of the patients, a >50% and >30% decline, respectively, of the serum tumor marker prostate specific antigen (PSA) was reached with a significant improved progression-free survival in patients with EGFR overexpression [28].

EGFR can not only serve as a target for antibodies or inhibitors, which are intended to downregulate EGFR-dependent signaling pathways. Since EGFR is internalized into the cell after ligand or antibody binding [29,30], it can also be used as a carrier for the targeted delivery of toxins that unfold their cytotoxicity inside the cancer cells. 

Various targeted toxins against EGFR were therefore generated in the past and tested against different hematological and solid tumors. The anti-EGFR mAbs cetuximab or panitumumab, anti-EGFR scFv thereof, TGFα, or EGF, were used as binding domains and enzymatic active domains of ribosome-inactivating proteins, like *Pseudomonas* Exotoxin A, Saporin, Dianthin, or Diphtheria toxin, were used as toxin domains (rev. in [31]). Yip and colleagues developed a conjugate consisting of the chimeric murine-human mAb cetuximab bound to Saporin by a biotin-streptavidin linker. Cytotoxicity against DU145 PCa cells was enhanced by photochemical internalization, leading to direct release of the conjugate from the endo-lysosomal compartment into the cytosol [32]. Targeted toxins consisting of the anti-EGFR scFv2112 from cetuximab or scFv1711 from panitumumab and the truncated version of *Pseudomonas* Exotoxin A (ETA’) were also tested against different tumor entities, including PCa. A high and specific cytotoxicity was determined on C4-2 PCa cells [29].

Due to the murine origin of their binding domains and the bacterial or plant origin of their toxin domains, targeted toxins are considered immunogenic in patients, which makes clinical use risky [33]. In the present study, we therefore generated new recombinant anti-EGFR targeted toxins for the treatment of PCa. We chose the natural human EGF ligand as binding domain and PE40, the *C*-terminal part of *Pseudomonas* Exotoxin A (PE), lacking the CD91 binding domain I and consisting of the domains II, Ib, and III with 40 kDa in size, as the toxin domain. Moreover, we generated a targeted toxin variant with EGF and a de-immunized PE domain, called PE24mut. In this toxic domain of 24 kDa in size, parts of domain II containing immunodominant B- and T-cell epitopes are deleted and only the furin cleavage site is retained. Moreover, seven immunodominant B-cell epitopes of domain III are mutated to alanines (R427A, R458A, D463A, R467A, R490A, R505A, R538A) [34]. We tested the targeted toxins EFG-PE40 and EGF-PE24mut on different PCa cell lines, representing advanced stages of the disease, in view of protein biosynthesis inhibition, cytotoxicity and induction of apoptosis. We found that both are promising candidates for further development to be used for the future treatment of PCa.

## 2. Results

### 2.1. Cloning, Expression and Purification of EGF and the Targeted Toxins EGF-PE40 and EGF-PE24mut

The natural EGF ligand and the targeted toxins EGF-PE40 and EGF-PE24mut were generated by cloning EGF via *NcoI/NotI* restriction sites into the vector pHOG21 containing a c-myc and a His-tag for detection and purification, followed by the insertion of the PE40 or PE24mut domains via *XbaI* restriction site (Figure 1). 

For expression and purification, vector DNA of EGF, EGF-PE40 and EGF-PE24mut was transformed into *E.coli* XL-1 blue bacteria and purified via immobilized metal affinity chromatography (IMAC) with a purity yield of approx. 95%, 82%, and 82%, respectively, as quantified by densiometric quantification of the SDS gels (Figure 2a,c,e). Western Blot analysis using anti c-myc antibody clearly identified the expression of the three proteins that appeared at their expected molecular masses in elution fractions, namely EGF at 8.4 kDa, EGF-PE40 at 49.2 kDa and EGF-PE24mut at 34.7 kDa (Figure 2b,d,f).

### 2.2. Binding of EGF-PE40 and EGF-PE24mut to Different PCa Cell Lines

EGFR expression was evaluated on the PCa cell lines LNCaP, DU145, and PC-3 by Western Blot using anti-EGFR rabbit pAb (Figure 3a). Abundant EGFR expression was found in all three PCa lines after flow cytometric analysis with 98.6% of positive population in LNCaP, 99.7% of positive population in DU145 and 96.5% of positive population in PC-3 cells (Figure 3b). 

Equilibrium dissociation constants (K_D_) of 3.8, 3.1, and 4.4 nM were determined for EGF on LNCaP, DU145, and PC-3 cells, respectively (Figure 3c, Appendix A). Compared to EGF, binding of EGF-PE40 was 28.6- to 76.1-fold reduced and binding of EGF-PE24mut was 6.0- to 11.9-fold reduced. No binding at saturated concentrations was seen on PSMA-negative CHO control cells.

### 2.3. EGF-PE40 and EGF-PE24mut Inhibit Protein Biosynthesis 

Since PE-based targeted toxins are known to inhibit the protein biosynthesis of target cells by ADP-ribosylation of eEF-2 [35], we checked whether there was an inhibition of protein biosynthesis in our cells lines after incubation with our constructs. Protein biosynthesis inhibition was analyzed using puromycin before lysis of the intoxicated cells. The antibiotic puromycin acts as an analog of the 3′aminoacyl-tRNA, causing the formation of puromycylated nascent chains [36,37]. The inhibition of protein biosynthesis can then be detected by Western Blotting using an anti-puromycin antibody. In all EGFR-positive PCa cell lines, a stronger protein biosynthesis inhibition with EGF-PE40 compared to EGF-PE24mut was found. Whereas EGF-PE40 led to nearly complete inhibition of protein biosynthesis after the mentioned time intervals in all PCa cell lines, EGF-PE24mut showed only partial protein biosynthesis inhibition in DU145 and PC-3 cells. No inhibition was observed in EGFR-negative CHO cells (Figure 4).

### 2.4. EGF-PE40 and EGF-PE24mut Reduce Cell Viability and Induce Apoptosis

Upon treatment of the PCa cells with EGF-PE40 or EGF-PE24mut, explicit morphological changes were observed in the PCa cells pointing towards cell death, whereas no signs of cell death or morphological changes were observed in CHO cells (Appendix A). Using WST-1 viability assay, we found that EGF-PE40 and EGF-PE24mut specifically reduced the viability of all PCa cell lines in a time and concentration dependent manner (Figure 5a, Table 1).

With EGF-PE40, IC_50_ values of 0.008 nM, 0.18 nM, and 0.86 nM were calculated for LNCaP, DU145, and PC-3 cells after 72 h incubation, respectively. For EGF-PE24mut, IC_50_ values of 0.97 nM, 5.27 nM, and 9.59 nM could be determined at the same time. Thus, they were about 11- to 120-fold lower compared to the IC_50_ values of EGF-PE40. No reduction in cell viability was observed in EGFR-negative CHO cells (Figure 5a, Table 1).

To evaluate the cause of cell death, PARP cleavage and Caspase-3 cleavage were checked as markers for apoptosis by Western Blot analysis. PARP and Caspase-3 cleavage were observed in LNCaP (48 h), DU145 (72 h) and PC-3 (72 h) cell lines after treatment with EGF-PE40 and EGF-PE24mut with tendency to a higher degree in EGF-PE40 treated cells. No PARP cleavage was observed in CHO cells (Figure 5b). In contrast to the EGF-targeted toxins, the EGFR inhibitor erlotinib showed no measurable cytotoxicity in the PCa cells up to a concentration of 5750 nM after 72 h (Table 1, Appendix A). Thus, EGF-PE40 was more than 6680 to 140,000-fold and EGF-PE24mut more than 600 to 3000-fold cytotoxic than erlotinib. Whole Western Blots including densitometry ratios are in Appendix A.

## 3. Discussion

We have successfully generated new targeted toxins that bind to EGFR and effectively induce apoptosis in androgen-dependent and independent growing PCa cells, which represent different stages of advanced disease [38].

Clinical studies with PE-based targeted toxins against various solid malignancies have shown formation of anti-drug antibodies (ADA) against the non-human binding domains and against the bacterial toxin domains leading to dangerous side effects and discontinuation of therapy [39]. The focus is therefore increasingly on the generation of targeted toxins with lower immunogenicity.

We used the natural ligand EGF as binding domain for our targeted toxins, which has the advantage over murine or chimeric anti-EGFR antibodies or scFv (scFv) [32,40] that it should not be immunogenic due to its human origin. As toxin domain, we used the naturally occurring cytotoxic domain of *Pseudomonas* Exotoxin A, PE40, and a de-immunized variant thereof, called PE24mut, which was shown to reduce the immunogenicity of the bacterial domain [34]. With the use of *Pseudomonas* Exotoxin A, we exploited the originally harmful properties of this virulence factor, which developed during evolution, for the therapy of cancer cells [41]. These properties include the formation of a furin cleavable motif within domain II for effective endosomal release of the toxin domain and a *N*-terminal KDEL-like motif for retrograde transport of the toxin domain via the Golgi apparatus to the ER. Furthermore, a translocation domain for passage through biological membranes has been formed as well as the ability to specifically ADP-ribosylate diphthamide, an amino acid that has so far only been described in eEF-2 [42].

We found that cell binding of EGF was reduced by addition of the PE24mut toxin domain and to an even greater extent by addition of the PE40 domain. It can thus be assumed that there was some steric inhibition of the EFG ligand by the toxin domains. The lower binding of EGF-PE40 compared to EGF-PE24mut, however, did not lead to reduced cytotoxic effects. Instead, EGF-PE40 led to a nearly complete inhibition of protein biosynthesis in the PCa cells and to a time-dependent cytotoxicity with IC_50_ values in the low nanomolar to picomolar range. EGF-PE24mut led to IC_50_ values in the single-digit nanomolar range and was thus significantly less toxic than EGF-PE40. As a cause we found a lower inhibition of protein biosynthesis by EGF-PE24mut compared to the PE40-based immunotoxin. Further investigations need to show if this is due to a reduced cytosolic uptake, since the translocation domain is completely deleted in PE24mut with exception of the furin cleavable site. Overall, the C4-2 and PC-3 cells proved to be less sensitive to the targeted toxins than the LNCaP cells. Not all cells of these lines were killed within a time period of 72 h and the cleavage of caspase-3 and PARP was not as pronounced as in LNCaP cells. Therefore, further investigations are needed to determine whether the targeted toxins did not enter the cytosol sufficiently due to increased lysosomal or proteasomal degradation or other obstacles during trafficking, or whether there is increased apoptosis resistance against the targeted toxins in these two cell lines.

The cytotoxicity of our targeted toxins are comparable with those of Niesen and colleagues, that consisted of the anti-EGFR scFv 2112 from Cetuximab or the anti-EGFR scFv 1711 from Panitumumab and ETA’, a truncated version of PE. Both showed a high cytotoxicity on the PCa cell line C4-2 with IC_50_ values of 55 pm for scFv2112-ETA‘ and 192 pM for scFv1711-ETA‘ after 72 h, respectively [29].

Despite a reduced cytotoxicity compared to our PE40-based immunotoxin, EGF-PE24mut should be more suitable for use in patients than EGF-PE40 due to its lack of immunogenicity. Moreover, it was still proven to be at least 600 to 3000-fold more cytotoxic than the EGFR inhibitor erlotinib, which mostly failed in clinical studies with patients suffering from PCa [24,26]. The increased cytotoxicity of our targeted toxins compared to inhibitors, like erlotinib, can be attributed to their enzymatic ADP-ribosyltransferase activity. Thus, only a few toxin molecules are sufficient to inhibit the protein biosynthesis of the target cells to such an extent that apoptosis is triggered. With competitive inhibitors, such as erlotinib, however, only a stoichiometric one to one binding can result in receptor tyrosine kinase blocking.

Since EGFR is also expressed on normal cells, it cannot be excluded that the treatment of PCa patients with our targeted toxins could lead to side effects due to on-target/off-tumor activities. It is therefore conceivable that in the future a combination therapy would be helpful with drugs that are directed against, for example PSMA, which is largely expressed on PCa cells and mostly organ-specific [43]. It appears to be reasonable to overcome the reduced efficiency of EGF-PE24mut because the minimized immunogenicity of this fusion protein is a substantial advantage for clinical applications. Testing of our targeted toxins in mouse models is required in the future with regard to reduced immunogenicity versus antitumor activity. Diminished cytosolic uptake due to the lack of the toxin’s intrinsic domain required to support membrane transfer might be compensated by strategies described to temporarily weaken the membrane integrity comprising lysosomotropic amines, carboxylic ionophores and calcium channel antagonists [44]. The most promising techniques in this regard include the use of cell-penetrating peptides [45], light-induced techniques [46] and co-application of glycosylated triterpenoids [47]. 

In summary, we could show that PE-based targeted toxins with the EGF ligand as binding domain are promising candidates for the treatment of prostate cancer. By using this human ligand and the de-immunized PE24mut toxin domain, the potential risk of immunogenicity in PCa patients could be diminished in the future.

## 4. Materials and Methods 

### 4.1. Cells and Chemicals

The EGFR-expressing human PCa cell lines LNCaP, DU145, ad PC-3 were obtained from the American Type Culture Collection (ATCC, Manassas, VA, USA). Cell line identity was verified by short tandem repeat (STR) analysis (CLS GmbH, Eppelheim, Germany). A Chinese hamster ovary (CHO) cell line with lack of human EGFR expression served as EGFR-negative control and was purchased from Gibco (Gibco, Invitrogen, Karlsruhe, Germany). LNCaP, DU145 and PC-3 cells were routinely propagated in complete RPMI 1640 medium, and CHO cells were propagated in complete F-12 medium (Gibco, Invitrogen, Karlsruhe, Germany), each with 10% fetal bovine serum (Biochrom, Berlin, Germany), penicillin (100 U/mL) and streptomycin (100 mg/L) as supplements, and incubated at 37 °C and 5% CO_2_ to maintain exponential cell growth and proliferation. Erlotinib HCl (Selleck Chemicals Llc, Houston, TX, USA) was dissolved in DMSO as a 4 mg/mL stock solution, aliquoted and stored at −80 °C until use.

### 4.2. Cloning, Expression and Purification of EGF and the Anti-EGFR Immunotoxins

The human EGF ligand and the immunotoxins EGF-PE40, and EGF-PE24mut were cloned into the expression vector pHOG21 followed by chemical transformation of XL-1 blue supercompetent *E. coli* cells (Agilent Technologies, Santa Clara, CA, USA). Correct gene sequences were verified by gene sequencing (Microsynth Seqlab, Göttingen, Deutschland). The vector pHOG21 contains a 22 amino acid leader peptide (pel-B leader) at the N-terminus that directs the protein to the bacterial periplasm [48]. Periplasmatic expression and purification of EGF, EGF-PE40 and EGF-PE24mut via immobilized affinity chromatography (IMAC) were carried out as described earlier [49]. In brief, one successfully transformed bacteria clone each was inoculated over night at 37 °C in YT medium containing 0.1 M glucose and 100 mg/mL ampicillin. The following day, the culture was diluted 1:20 and grown in 600 mL medium until reaching an OD of 0.8, then pelleted by centrifugation and resuspended in 600 mL YT medium containing 50 mg/mL ampicillin, 0.4 M sucrose and 1 mM IPTG for overnight protein expression at RT. Bacteria cultures were centrifuged (6000 rpm, 4 °C, 10 min) and pellets were resuspended in 30 mL ice-cold Tris-HCl, 20% sucrose, 1 mM EDTA (pH 8.0) and incubated on ice for 1 h for the release of periplasmic proteins. For harvesting the periplasmatic extract, the suspension was centrifuged (10,000 rpm, 4 °C, 30 min) and supernatant was collected and dialyzed against 50 mM Tris-HCl, 1 M NaCl (pH 7.0). The periplasmatic extract (PE) was then purified with the help of HiTrap^TM^ Chelating High Performance columns with chelating sepharose (Sigma-Aldrich, St. Louis, MO, USA) charged with Ni^2+^ metal ions. Polyhistidine tagged EGF and immunotoxins were purified subsequently: The column was equilibrated with equilibration buffer [50 mM Tris-HCl, 1 M NaCl (pH 7.0)] and PE was loaded. Bound target proteins were eluted stepwise with varying imidazole concentrations of 40 mM (elution fraction E1), 80 mM (E2), and 250 mM (E3-5) and dialyzed against phosphate-buffered saline (PBS). Final protein yield of all preparations was determined by NanoDrop^TM^ Lite Spectrophotometer (Thermo Fisher Scientific, Waltham, MA, USA). 

### 4.3. Preparation of Cell Lysates 

Target cells were lysed in RIPA buffer (50 nM Tris-HCl, 150 nM NaCl, 1 mM EDTA, 0.5% NaDeoxycholate, 0.05% SDS, 1% Igepal), supplemented with Protease Inhibitor Cocktail (Merck Millipore, Burlington, MA, USA) and Phosphatase Inhibitor Cocktail (Sigma-Aldrich, St. Louis, MO, USA), and incubated on ice for 30 min. After centrifugation (13,000 rpm, 4 °C, 30 min), the supernatant was collected and quantification of protein content was determined by Quick Start^TM^ Bradford Protein Assay (Bio-Rad Laboratories, Inc., Hercules, CA, USA). The whole cell lysates were aliquoted and stored at −80 °C.

### 4.4. SDS-PAGE and Western Blot 

SDS-PAGE and Western Blot were used to verify the expression and purification of EGF, EGF-PE40, and EGF-PE24mut according to previous descriptions [50]. Purified preparations were detected by HRP-conjugated c-myc mouse mAb (Roche Diagnostics, Mannheim, Germany, #11814150001). EGFR expression in the target cells was assessed by Western Blotting of whole cell lysate (50 µg per lane) with the help of EGFR rabbit pAb (Santa Cruz Biotechnology, Dallas, TX, USA, #sc-03) and HRP-labeled goat anti-rabbit pAb (Dako Denmark A/S, Glostrup, Denmark, #P0448). 

Induction of apoptosis in the target cells was determined by Western Blot (50 µg per lane). Caspase-3 activation and poly-(ADP-ribose)polymerase (PARP) cleavage were detected by Cas-3 mouse mAb (ECM Biosciences, Versailles, KY, USA, #CM4911) and PARP rabbit pAb (Cell Signaling Technology Europe, Leiden, The Netherlands, #9542) as primary antibodies. HRP-labeled goat anti-rabbit pAb (Dako Denmark A/S, Glostrup, Denmark, #P0448) and HRP-labeled rabbit anti-mouse pAb (Dako Denmark A/S, Glostrup, Denmark, #P016) served as secondary antibodies. 

β-Actin was used as a loading control and detected by HRP-conjugated β-Actin rabbit pAb (Cell Signaling Technology Europe, Leiden, The Netherlands, #5125). Visualization and analysis of protein bands was carried out by developing the membranes with an enhanced chemiluminescence (ECL) system (Clarity^TM^ Western ECL Substrate, Bio-Rad Laboratories, Inc., Hercules, CA, USA), the ChemiDoc^TM^ MP Imaging System and the software Image Lab^TM^.

### 4.5. Flow Cytometry

Characterization of EGFR expression on the target cell surface was examined by flow cytometric analysis as described by us earlier [50]. PE-labeled EGFR mAb (BioLegend, San Diego, CA, USA, #352903) was used as detection antibody.

Binding of the EGF ligand, EGF-PE40, and EGF_PE24mut to EGFR on the target cells was confirmed by flow cytometry. Bound EGF and targeted toxins were detected via His-tag rabbit mAb (Cell Signaling Technology Europe, Leiden, The Netherlands, #12698) and goat anti-rabbit IgG-PE (Southern Biotech, Birmingham, AL, USA, #4010-09S). Mean fluorescence values of stained cells were determined with a FACS Calibur flow cytometer and the software CellQuest Pro (BD Biosciences, Heidelberg, Germany).

### 4.6. Inhibition of Protein Biosynthesis

For the verification of protein biosynthesis inhibition via Western Blot, cells were incubated with EGF-PE40 or EGF-PE24 at approx. IC_20_ concentrations of EGF-PE40 for 48 h (LNCaP) or 72 h (DU145, PC-3, CHO). Puromycin (5 µg/mL) (Tocris, Bio-Techne GmbH, Wiesbaden, Germany) was added to the cells 15 min before harvesting. RIPA lysate preparation was carried out as described above. Western Blot membranes were stained with puromycin mouse mAb (Merck Chemicals, Darmstadt, Germany, #MABE343) and HRP-labeled rabbit anti-mouse pAb (Dako Denmark A/S, Glostrup, Denmark, #P016). 

### 4.7. WST-1 Cell Viability Assay 

Cytotoxicity after immunotoxin treatment was quantified by the colorimetric WST-1-Assay according to the standard protocol (Roche Diagnostics). Based on the enzymatic cleavage of the water-soluble tetrazolium salt WST-1 (2–(4–Iodophenyl)–3–(4–nitrophenyl)–5-(2,4–disulfophenyl)–2H–tetrazolium, monosodium salt) to formazan, the number of metabolically active cells can be approximated proportionally. 

### 4.8. Bright-Field Microscopy

Morphology of the target cells after treatment with EGF-PE40 and EGF-PE24mut was studied by bright-field microscopy using the confocal laser scanning microscope LSM 880 with Airyscan (Carl Zeiss Microscopy, Göttingen, Germany).

### 4.9. Statistical Analysis

Binding affinity of EGF was calculated by GraphPad Prism 7 software (GraphPad Software, San Diego, CA, USA). K_D_ values were defined as the drug concentrations leading to half-maximal specific binding. 50% inhibitory concentrations (IC_50_) values of WST-1 data were calculated for the targeted toxins and erlotinib on each cell line using GraphPad Prism 7 software by non-linear regression ([inhibitor] vs. normalized response).

## Figures and Tables

**Figure 1 toxins-12-00753-f001:**
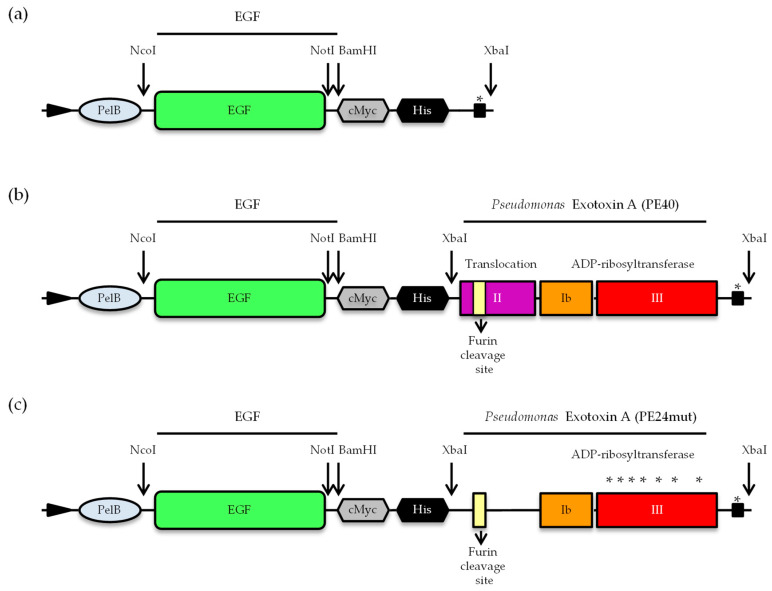
Cloning of EGF and the targeted toxins EGF-PE40 and EGF-PE24mut. Schematic representation of (**a**) EGF, (**b**), EGF-PE40, and (**c**) EFG-PE24mut in the vector pHOG21. (Abbreviations: *c-myc*, human c-myc tag; *His_6_*, hexahistidine tag; *PelB*, pel B leader for periplasmatic expression. * Mutations of amino acids within immunodominant epitopes to alanine for de-immunization.

**Figure 2 toxins-12-00753-f002:**
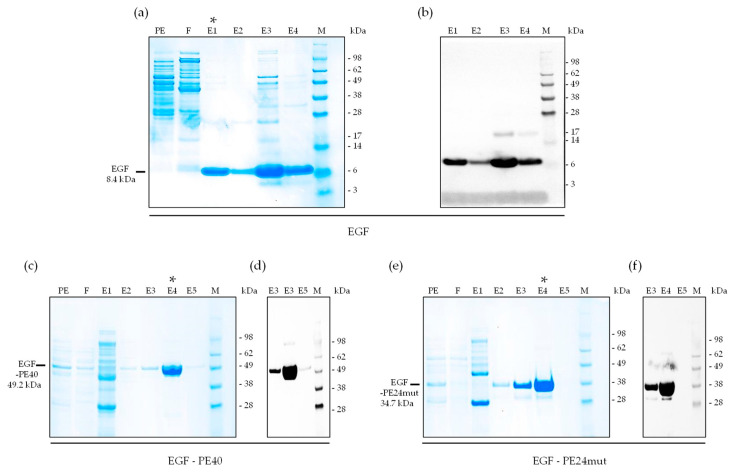
Expression and purification of EGF, EGF-PE40, and EGF-PE24mut. (**a**) SDS-PAGE and (**b**) Western Blot of purified EGF found in the elution fractions E1-E4. (**c**) SDS-PAGE and (**d**) Western Blot of purified EGF-PE40 found in the elution fractions E2-E4. (**e**) SDS-PAGE and (**f**) Western Blot of purified EGF-PE24mut found in the elution fractions E2-E4. * elution fraction with highest purity used for the following experiments. Abbreviations: E1–4, elution fractions 1–4; F, flow-through; M, marker; PE, periplasmatic extract.

**Figure 3 toxins-12-00753-f003:**
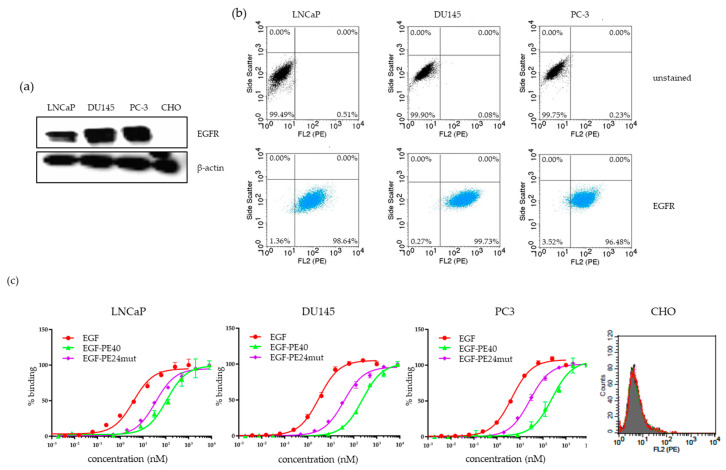
EGFR expression and EGF binding to different PCa cell lines. (**a**) EGFR expression on different PCa cell lines (LNCaP, DU145, PC-3) and CHO control cells as shown by Western Blot. (**b**) Flow cytometric analysis of EGFR-positive cell populations in LNCaP, DU145, PC-3 and EGFR-negative CHO cells. (**c**) Binding of EGF, EGF-PE40, and EGF-PE24mut to PCa cells at saturated concentration on PSMA-negative CHO cells.

**Figure 4 toxins-12-00753-f004:**
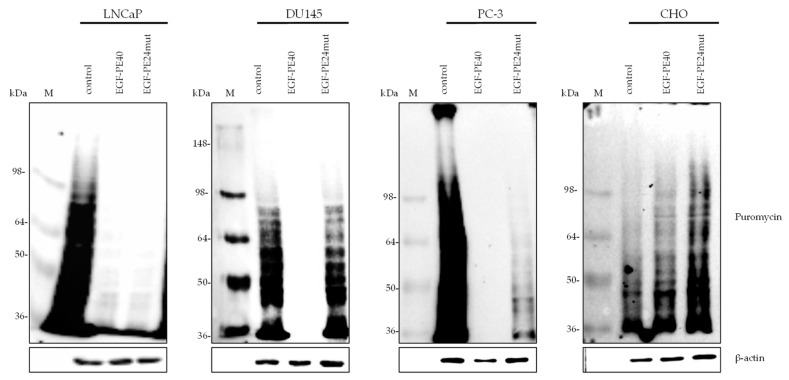
Protein biosynthesis inhibition in PCa as shown by puromycin Western Blot. EGF-PE40 and EGF-PE24mut inhibited protein biosynthesis of LNCaP cells after 48 h incubation with 0.7 nM EGF-PE40 or EGF-PE24mut, of DU145 cells after 72 h incubation with 1 nM of the targeted toxins, and of PC-3 as well as CHO cells after 72 h incubation with 2 nM of the targeted toxins.

**Figure 5 toxins-12-00753-f005:**
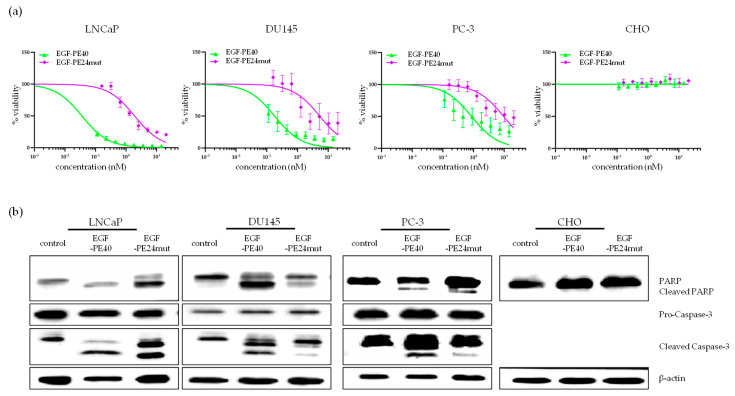
EGF targeted toxins reduce cell viability and induce apoptosis. (**a**) EGFR-positive LNCaP, DU145, PC-3 and EGFR-negative CHO cells were incubated with EGF-PE40 or EGF-PE24mut for 48 (LNCaP) or 72 h (DU145, PC-3 and CHO cells), respectively. Reduction of cell viability was calculated by WST-1 assay. Mean values ± SEM of three independent experiments. (**b**) Western Blots of the apoptotic markers PARP, cleaved PARP, Caspase-3 and cleaved Caspase-3 after incubation of the cells with EGF-PE40 or EGF-PE24mut at the same time points. β-actin was used as loading control.

**Table 1 toxins-12-00753-t001:** IC_50_ values of EGF-PE40, EGF-PE24mut, and erlotinib on PCa cells. CHO cells served as control. IC_50_ values, defined as half-maximal inhibitory concentrations, were determined using WST-1 viability assay and GraphPad Prism 7 software.

	LNCaP			DU145			PC-3			CHO		
	IC_50_(nM)			IC_50_(nM)			IC_50_(nM)			IC_50_(nM)		
	24h	48h	72h	24h	48h	72h	24h	48h	72h	24h	48h	72h
EGF-PE40	2.31	0.04	0.008	2.71	0.31	0.18	9.67	0.64	0.86	>14.5	>14.5	>14.5
EGF-PE24mut	>20.6	1.91	0.97	>20.6	8.52	5.27	>20.6	8.80	9.59	>20.6	>20.6	>20.6
Erlotinib	nd	>5750	nd	nd	nd	>5750	nd	nd	>5750	nd	nd	>5750

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
