# Peer review of "Pseudomonas Exotoxin A Based Toxins Targeting Epidermal Growth Factor Receptor for the Treatment of Prostate Cancer"

_toxins, 2020, doi:10.3390/toxins12120753_

Round 1
Reviewer 1 Report
The authors made EGF-PE fusion protein and evaluated the cytotoxic activity against prostate cancer lines. They used PE40 and PE24mut molecules as toxin moieties which were developed by NCI/NIH.
EGF targeted immunotoxins were studied by many researchers and evaluated in vitro activity using colon cancer and/or prostate cancer etc. PE40 and PE24mut molecule have been developed by NCI/NIH researchers and these molecules has been evaluating in clinical trials. For the scientific reasons, in vivo data is needed. However, this manuscript does not have in vivo data. If they could present in vivo data, scientific merit is big and clinical researcher would be satisfied.
Minor point
Line 82-85; There is an error of the mutation residues in PE24mut. Also there is an error in citation. As PE24mut was discovered by NCI/NIH group, citation should be their PNAS paper.
Figure 5; Citotoxicity curve should be fixed, because some curves are under 50% viability at minimum dose.
Author Response
We thank for the helpful comments.
Our point by point answers are as follows:
- EGF targeted immunotoxins were studied by many researchers and evaluated in vitro activity using colon cancer and/or prostate cancer etc. PE40 and PE24mut molecule have been developed by NCI/NIH researchers and these molecules has been evaluating in clinical trials. For the scientific reasons, in vivo data is needed. However, this manuscript does not have in vivo data. If they could present in vivo data, scientific merit is big and clinical researcher would be satisfied.
We fully agree with the reviewer that animal experiments are required to estimate the final clinical potential of the targeted toxins: we have also mentioned it in the manuscript at page 8, lines 236-237. However, this was not the scope of the present manuscript. Here, we wanted to demonstrate as proof of principle that targeted toxins containing EGF as binding domain show high and specific cytotoxicity against EGFR expressing prostate cancer cells. This is in agreement with the comments of reviewer 2 and 3 who both consider the present study to be conclusive and completed. Moreover, the editor provided a deadline of one week for the corrections, which implicates that the editor follows the opinion of reviewer 2 and 3 as it is obvious that animal experiments would take several months of time.
Minor point
- Line 82-85; There is an error of the mutation residues in PE24mut. Also there is an error in citation. As PE24mut was discovered by NCI/NIH group, citation should be their PNAS paper.
The error in the mutation residues was corrected. The PNAS paper was included as reference (now lines 85-86).
- Figure 5; Citotoxicity curve should be fixed, because some curves are under 50% viability at minimum dose.
The toxicity curves were modified in a new Fig. 5 (a) to better reflect the cytotoxicity.
Reviewer 2 Report
The paper "Pseudomonas Exotoxin A Based Toxins Targeting 3 Epidermal Growth Factor Receptor for the Treatment 4 of Prostate Cancer" describes the properties of novel targeted toxins against different PC cell lines. The paper is interesting and I suggest its publication with minor changes.
pag. 1 row 6: the Authors should substitute 'The epidermal growth factor’ with ‘The epidermal growth factor receptor’
pag. 2 row 56: the Authors should indicate the meaning of PSA
Pag. 5 rows 134-135: please add a reference about puromycin functioning
Pag.5 rows 139-140: “EGF-PE24mut showed weaker inhibition with only small extend in DU145 cells”: please reformulate this phrase in a clearer way: do you mean that the effect in DU145 is very weak?
Pag. 6 figure 5: the hill coefficient in the fitting of red curves in DU145 and PC-3 cells seems higher than 1: do you have any comments for this evidence? The viability seems not going to zero even at higher concentration of the toxins (also for the green curves): do you have any comments on this?
Pag 7 row 162: change “nm” for “nM”
Pag 7 row 182: please add a reference after: “over some anti-EGFR single chain variable fragments (scFv)”
Pag 7 rows 182-183: clarify or reformulate this phrase “that it is not to be immunogenic”: do you mean “that should not be immunogenic”?
Pag 8 row 202: please specify that ETA means: a truncated version of Pseudomonas exotoxin A.
pag. 9 rows 249-250: “and pellets were resuspended in 250 mL ice-cold Tris-HCl, 20 % sucrose, 1 mM EDTA (pH 8.0) and incubated on ice for 1 h” is this the procedure to collect periplasmic extract? Please specify or add a reference. In addition, please explain why the targeted toxins are localized in the periplasm of e. coli: I have found this information just in Fig. 1 “pel B leader for periplasmatic expression” you should add some explanation or reference to explain periplasm expression in the method section.
Author Response
We thank for the helpful comments.
Our point by point answers are as follows:
- 1 row 6: the Authors should substitute 'The epidermal growth factor’ with ‘The epidermal growth factor receptor’
The sentence was corrected.
- 2 row 56: the Authors should indicate the meaning of PSA
The meaning of PSA was indicated.
- 5 rows 134-135: please add a reference about puromycin functioning
The references of Yarmolinski and Haba, 1959 [34] and of Nathans, 1964 [35] were added (now row 141)
- 5 rows 139-140: “EGF-PE24mut showed weaker inhibition with only small extend in DU145 cells”: please reformulate this phrase in a clearer way: do you mean that the effect in DU145 is very weak?
The sentence was reformulated with clearer statements (rows 145-147).
- 6 figure 5: the hill coefficient in the fitting of red curves in DU145 and PC-3 cells seems higher than 1: do you have any comments for this evidence? The viability seems not going to zero even at higher concentration of the toxins (also for the green curves): do you have any comments on this?
Fig. 3 (c) was exchanged with binding curves for EGF, EGF-PE40 and EGF-PE24mut on the different cell lines to demonstrate how the binding of the EGF ligand changed when the toxin domains were added. KD values of the binding curves were added in a new table in the supplement (Supplementary table 1). Binding was described on page 4, rows 114-126, and discussed at pages 7 and 8, rows 199-203. The method section was revised at page 10, rows 311-312.
Our goal was to show the efficiency of EGF- PE40 and EGF-PE24 mut by comparison at different points of time. The fact that the viability does not seem to go to zero in some cases was discussed in more detail at page 8 lines 209-215.
- Page 7, row 162: change “nm” for “nM”
nm was changed to nM (row 169).
- Page 7, row 182: please add a reference after: “over some anti-EGFR single chain variable fragments (scFv)”
The reference of Azemar et al., 2000 [36] was added (row 189).
- Page 7, rows 182-183: clarify or reformulate this phrase “that it is not to be immunogenic”: do you mean “that should not be immunogenic”?
The sentence was corrected (row 189).
- Page 8, row 202: please specify that ETA means: a truncated version of Pseudomonas exotoxin A.
The sentence was reformulated (row 218).
- 9 rows 249-250: “and pellets were resuspended in 250 mL ice-cold Tris-HCl, 20 % sucrose, 1 mM EDTA (pH 8.0) and incubated on ice for 1 h” is this the procedure to collect periplasmic extract? Please specify or add a reference. In addition, please explain why the targeted toxins are localized in the periplasm of e. coli: I have found this information just in Fig. 1 “pel B leader for periplasmatic expression” you should add some explanation or reference to explain periplasm expression in the method section.
More explanations were given about the pelB leader (rows 260-262) and the production of the periplasmatic extract (row 270). The reference of Kipriyanov et al., 1997 [43] was added.
Reviewer 3 Report
In this work, Pseudomonas Exotoxin A (PE40) is suggested as potential therapy for prostate cancer. The suggested toxin has affinity to EGFR in well-stablished prostate cancer cell lines such as LNCaP, PC-3 and DU145. A de-immunized variant of PE40 had been used as a control (PE24mut).
In my review, this manuscript is well-written and the data are presented clearly. The provided supplemental information are helpful. Therefore, I wish to recommend this manuscript to be considered for publication.
Author Response
No issues were raised.
We thank for reviewing the manuscript.